# Development of a Novel, Automated High-Throughput Device for Performing the Comet Assay

**DOI:** 10.3390/ijms24087187

**Published:** 2023-04-13

**Authors:** Mahsa Karbaschi, Yunhee Ji, Mubarak A. Mujawar, Mario Mendoza, Juan S. Marquez, Apurva Sonawane, Pratikkumar Shah, Chris Ross, Shekhar Bhansali, Marcus S. Cooke

**Affiliations:** 1Independent Researcher, Sunnyvale, CA 94089, USA; 2Oxidative Stress Group, Department of Molecular Biosciences, University of South Florida, Tampa, FL 33620, USA; 3Department of Electrical and Computer Engineering, Florida International University, Miami, FL 33174, USA; mmujawar@fiu.edu (M.A.M.); mmendoza@fiu.edu (M.M.); jmarq056@fiu.edu (J.S.M.); asona003@fiu.edu (A.S.); pratikkumar.shah@fiu.edu (P.S.);; 4Engineering Resources Group, Pembroke Pines, FL 33029, USA; chris.ross@engr-group.com

**Keywords:** high throughput, automated, comet assay, single-cell gel electrophoresis, DNA damage, DNA repair, oxidative stress, genotoxicity

## Abstract

A comet assay is a trusted and widely used method for assessing DNA damage in individual eukaryotic cells. However, it is time-consuming and requires extensive monitoring and sample manipulation by the user. This limits the throughput of the assay, increases the risk of errors, and contributes to intra- and inter-laboratory variability. Here, we describe the development of a device which automates high throughput sample processing for a comet assay. This device is based upon our patented, high throughput, vertical comet assay electrophoresis tank, and incorporates our novel, patented combination of assay fluidics, temperature control, and a sliding electrophoresis tank to facilitate sample loading and removal. Additionally, we demonstrated that the automated device performs at least as well as our “manual” high throughput system, but with all the advantages of a fully “walkaway” device, such as a decreased need for human involvement and a decreased assay run time. Our automated device represents a valuable, high throughput approach for reliably assessing DNA damage with the minimal operator involvement, particularly if combined with the automated analysis of comets.

## 1. Introduction

Genomic instability is a hallmark of many major human diseases, such as cancer, cardiovascular disease, and neurodegenerative diseases [1,2,3,4]. Therefore, effective assessment of cellular DNA damage and repair could facilitate disease prevention, biomarker development, biomonitoring of environmental contamination/exposure, screening of individuals at a high risk of cancer, toxicity testing of the drugs and new therapeutic/diagnostic approaches, and the discovery of novel therapeutic methods. The sensitivity detection and quantification of DNA modifications is an essential requirement for the analysis of DNA damage, cellular DNA repair activity, and antioxidant capacity. Whilst many techniques have been developed for this purpose, single-cell gel electrophoresis (comet assay) continues to increase in popularity [5], not least because it is a simple method for studying the formation of a wide variety of forms of nuclear DNA damage, both in vitro and in vivo, as a marker of genotoxicity [6] and is widely applicable to numerous different cell types, ranging from yeast to human cell types [5]. An assay has applications in both academia and the pharmaceutical industry, for drug genotoxicity screening, monitoring environmental genotoxicity hazards, molecular epidemiology, and fundamental research in DNA damage and repair [6,7,8,9,10]. Compared to the other available methodologies for the detection of DNA damage, a comet assay is relatively inexpensive and highly sensitive [6]. While there have been some significant attempts to improve inter-laboratory agreement in the levels of damage measured, largely driven by the European Comet Assay Validation Group [11,12,13], and some new applications, e.g., the assessment of DNA damage in whole blood [14,15,16], the core comet assay protocol has remained largely unchanged since it was originally described by Ostling & Johansson [17] and Singh et al. [18]. However, an assay’s relatively low sample throughput, questions regarding inter-laboratory reproducibility, laborious multi-step sample workup procedure (Figure 1), and the time-consuming image analysis are all possible limiting factors for its wider adoption. For these reasons, several high-throughput methods have been developed [19,20,21] to improve and expedite the processing of larger numbers of samples than conventional methods [19,20,22,23]. Although these innovations have increased sample throughput for a comet assay, in terms of the numbers of samples handled, it remains time-consuming (up to three days) and, for the most part, labor-intensive (multiple steps, multiple slides of samples, which are processed individually, Figure 1).

Earlier, we developed a high-throughput method for processing comet assay slides (known as the HT comet assay) [23], which exploits the novel, vertical orientation of slides in a slide carrier. This approach increases the number of slides processed and decreases the number of individual slide manipulations, the footprint of the tank, the reagent requirements, and the time spent manipulating the slides. However, this method, as with all comet assay methods, still requires user involvement at numerous key steps, which makes it less user-friendly, and hence may limit wider adoption. 

In this brief report, for the first time, we describe the development of a device that exploits the vertical slide orientation of our HT comet assay approach and automates the entire process of a comet assay from the lysis step up to scoring (Figure 1). We termed this the automated high-throughput comet assay device (AHT device), and subsequently tested its performance against our HT comet assay. The AHT comet assay device removes the need for repeated human intervention throughout the process, and provides consistent and accurate temperature control, as well as the prevention of light contamination, which is thought to introduce artefactual damage. In addition, this device is a fully walkaway system, which decreases the assay run time from three days to under 5 h, removing all slide manipulations, and most operator involvement. Combined, this leads to a more user-friendly process, while allowing users to be more efficient by relinquishing their need to supervise and maintain constant attendance in the assay, which can be a long and tedious process.

## 2. Results

We have developed a novel device to automatically process conventional comet assay microscope slides with cell-containing gels, throughout the comet assay process (illustrated by Figure 1). The device is described and shown in the Material and Methods section. The device was first tested to determine whether it could successfully replicate the conditions and steps of the comet assay process. We then aimed to determine whether the AHT comet assay approach could be used successfully to automate comet assay slide processing. Finally, we compared the performance of the AHT comet assay system with the manual HT comet assay approach across an H_2_O_2_ dose response in cultured cells.

### 2.1. Initial Testing of the AHT Device

The initial testing of our novel AHT device demonstrated that comet assay fluids were successfully chilled to, and maintained at, 4 °C. Subsequently, we established that the correct voltage could be produced and maintained in an electrophoresis buffer in the electrophoresis tank. Finally, we confirmed that the device replicated the correct conditions of the comet assay procedure, in the correct steps, in the required sequence, and for the correct duration, as evidenced by the successful comet formation (see Figure 2, comet formed using the AHT device). This was achieved within 5 h, during which the operator was only required to start loading the slides into the device prior to the lysis step (equivalent to step III, Figure 1), and then return to collect them after final washing (equivalent to step XII, Figure 1) prior to scoring and analysis. This was a significant benefit for the operator, who was free to perform other tasks without having to return to the assay for step IV—XI.

### 2.2. Qualitative Comparison of Comets Generated by the AHT Device and the HT Assay

The HaCaT cells were treated with 0, 50, 75, 100, or 150 µM of H_2_O_2_ for 30 min on ice, and then the alkaline comet assay was performed using both the AHT device and the HT assay in parallel, using the same stocks of all buffers, on the same day. The quality of the comets (e.g., shape and uniformity) were then compared between the slides run by the HT assay or AHT device. The results showed an equivalent size, shape, uniformity, and staining intensity of the comets, together with a lack of background staining, in the samples run by both approaches (Figure 2).

### 2.3. Correlation between Comets Generated by the AHT Device and the HT Assay

A clear linear, dose-dependent increase in the level of DNA damage was achieved with the samples run on both the AHT device, and HT assay. For the samples treated with 0–75 µM H_2_O_2_, the mean value did not differ significantly between the two assay approaches (*p* > 0.05). However, at higher concentrations of H_2_O_2_ (100–150 µM), and hence levels of DNA damage, the samples run on the AHT device showed a lower mean percentage tail DNA (4.13% less tail DNA with 100 µM H_2_O_2_, and 4.3% less tail DNA with 150 µM H_2_O_2_, *p* < 0.05), compared to the HT assay (Figure 3A). At 150 µM H_2_O_2_, the dose response began to plateau, as shown by the comets generated by both the AHT device and the HT assay. A trend was noted for the results from the AHT device to have smaller error bars than the HT assay, suggesting less variability arising from processing samples with the AHT device. A linear regressing analysis was also performed on the data from the HT assay and AHT device (Figure 3B), which demonstrated a highly significant correlation coefficient close to one (*R*^2^ = 0.998, *p* < 0.0001). These results clearly demonstrated the feasibility of assessing DNA damage using our novel AHT device. Furthermore, at higher levels of DNA damage, the use of the AHT device potentially leads to less variability in the comet %tail DNA than the HT assay.

## 3. Discussion

The AHT comet assay device was developed with the aim of overcoming two key limitations of the existing comet assay procedures: (1) decreasing the total assay time to less than one day; and, perhaps more importantly, (2) removing the need for regular user involvement and attendance during the long, and sometimes tedious process, which improved its ease of use. Here, we report on the successful development and testing of the AHT device and its comparison with our well-established HT comet assay method. In the HT system, the slides are held in a vertical orientation in slide racks rather than in a horizontal orientation, as is the convention, which decreases the slide handling time by 60%, increases the number of slides per experiment, and decreases both bench space and reagent requirements. Although the HT assay does have increased sample throughput, three major limitations remain: the long procedure time, the need for multiple sample manipulations, and the repeated monitoring of the process. To develop a fully automated, walkaway system, the HT vertical electrophoresis tank was integrated into a newly designed automated liquid handling device to fully automate the comet assay slide processing procedure. An outline of the components of the device is shown in Figure 4. Hence, since the actual electrophoresis tank was the same as the HT tank, all the steps in the automated system reflected the optimized protocol for the HT system, e.g., duration of lysis, and electrophoresis. Therefore, the controlling software and hardware were assembled in such a way that they would replicate the incubation period, the buffer volumes, and the temperature requirements of all steps (washing/neutralization/draining/drying/rehydration/staining up to scoring) as in the optimized protocol of the HT system.

The key difference between the HT assay and the AHT device is that in the former method, the slides are placed in portable slide racks before the first step and then the racks containing the slides are then transferred between different treatment dishes, containing the appropriate solution at the required temperature for each step. In contrast, in the AHT device, the slide-containing racks are fixed vertically in place inside the electrophoresis tank during the entire assay, and the slides remain in the racks within the electrophoresis tank to which the solutions are subsequently added/removed, and electrophoresis performed. Specifically, as the process proceeds, the valves under the solution tanks open and the liquids flow into a custom-made cooling tank. Once the liquid is chilled to the appropriate temperature, the cooling tank is opened and the chilled liquid flows into the HT tank; therefore, the racks containing the slides are not removed during the assay run in the AHT device, and monitoring of the assay is not required by the user during the run. This key feature removes the risk of damage to the gels that exists in the conventional comet assay. Furthermore, the built-in chilling system in the AHT device obviates the need for an external fridge or chiller for cooling the buffers, or wet ice, as is the convention.

The results obtained from the HT assay and the AHT device showed highly comparable results in terms of the shape of the comets and linearity of the dose-response. Although the percentage tail DNA after the treatment with 100 or 150 µM H_2_O_2_ was lower in the comets processed by the AHT device, the mean value was only 4% lower than the comparable mean values from the slides processed in the HT assay. However, the error bars on the results in the AHT device were notably narrower than the results from the HT approach, demonstrating a lower dispersity in the percentage tail DNA across the comets.

The AHT comet system can be run overnight without any supervision, which is very convenient for the operator, allowing them to perform other tasks. Furthermore, the total assay run time using the AHT device is significantly shorter than the HT assay (less than 5 h vs. one day). The shorter assay run time offered by the AHT device provides the opportunity to increase the number of samples analyzed per day by running the device consecutively, which could be a significant benefit for the pharmaceutical industry, for example, where throughput is a critical factor. Throughput would be further increased in the HT electrophoresis tank used in the AHT device, which was to be combined with an approach such as the CometChip [24] (for which we have proof of principle (Ji et al. unpublished data [25]). Additionally, the AHT device offers substantial cost-savings in terms of the requirement to hire or train highly skilled operators (plus their time), a refrigerator, and an incubator, together with a need for laboratory space with sinks for washing, and a significant amount of dedicated bench space to run all the steps of a comet assay. The AHT device can provide other advantages to the comet assay such as precise and consistent temperature control the prevention of light exposure throughout the process (with the concomitant risk of artefactual DNA damage) and obviate the need for additional chilling (icemaker) and heating (incubator) of the device. The apparently increased consistency in the assay conditions and lack of human involvement offered by AHT device could not only increase the reproducibility of the data but also minimize intra- and inter- laboratory variation in the comet assay data.

It could be argued that the use of an AHT device is not compatible with an enzyme-modified comet assay. However, transferring the slides from an AHT device to incubate them with DNA repair enzyme is not precluded with this device. Although at this time, the enzyme incubation step stops the AHT device from being a fully walkaway device, it will still make a significant difference in the need for the operator involvement in subsequent steps, together with decreasing the assay footprint and the total assay run time, as noted above. Furthermore, a considerable number of laboratories do not use an enzyme-modified comet assay, and for these, this approach will represent a particularly welcome advance.

Our automated device represents a valuable, high throughput approach for reliably assessing DNA damage with minimal operator involvement. We propose that when used in conjunction with an automated scoring system (e.g., a Cytation 5, [26]) offers the potential for the comet assay to become even more rapid and a truly high throughput.

## 4. Materials and Methods

### 4.1. Overview of the Automated High-Throughput (AHT) Comet Assay Device

The AHT device described here was able to perform the entire comet slide processing steps following the push of a button (Figure 4). The device is comprised of three levels (Figure 5A). The top level consists of (1) the command center; tanks containing assay solutions (lysis buffer, electrophoresis buffer, neutralization buffer, and staining) and a water washing tank, each with its own solenoid valves; (2) a cooling chamber and first cooling loop; and (3) a sliding electrophoresis tank for easy loading and second cooling loop. As the process begins, pre-programmed, individual solenoids are turned on at different stages of the procedure allowing the solution’s transfer into the cooling chamber, where a float sensor determines the amount of fluid to be cooled. The middle level comprises a custom-made, insulated cooling module, containing a cooling loop made up of liquid cold plates (Wakefield-Vette, Pelham, NH, USA) connected to a high-flow liquid pump (SEAFLO, Glendale Heights, IL, USA), a CPU liquid cooler (Corsair, Freemont, CA, USA) to dissipate heat and, at the core of the device, a Peltier cooler (Hebei IT, Shanghai, China), which provides both cooling and heating. The cooling box made up of high-density polyethylene is packed with two float sensors (Cynergy3, Garden Groove, CA, USA) and a temperature sensor (Maxim Integrated Products, Sunnyvale, CA, USA). Once the liquid is cooled to the required temperature of 4–8 °C (which takes about 30 min), which is predefined for each solution and monitored by the Arduino microcontroller, the sixth solenoid at the base of the cooling chamber is opened, allowing its contents to be transferred into the electrophoresis tank (Cleaver Scientific Ltd., Warwickshire, UK), which is located in the middle level. Another cooling loop constantly chills and circulates the water around the electrophoresis tank to maintain its contents at 4 °C. All the electronics and power supply also rest on the top level for easy access, as required.

### 4.2. Technical Details of the AHT Device

The AHT device has a footprint of 48 cm by 48 cm and a height of 90 cm. The device consists of a stainless-steel frame, which supports three levels, each manufactured from approximately 0.5 cm thick fiberglass (Figure 5B). The top level has four 550 mL plastic chemical storage tanks and one 5.6 L spun aluminum water tank—all equipped with electrically-controlled solenoid outlet valves. A 10 cm segment of transparent, medical grade, PVC tubing (1/2-inch OD) descends from each solenoid valve into an 800 mL acrylic cooling chamber located on the second level. The state of the solenoid valves depends on the feedback from the two float sensors residing in this chamber. Together with the cooling chamber, the second level contains a 4.3 lpm/35 psi water pump, a 120 W Peltier cooler, and a 12 V brushless fan. The fluid entering the chamber is pumped through a Peltier device, which is cooled to the appropriate temperature. A thermocouple in the cooling chamber sends continuous temperature data to the device’s microcontroller to ensure accurate and consistent temperature control. The cooled fluid is then transferred to the electrophoresis tank on the lowest level, under gravity, via a solenoid valve-gated outlet. The modified electrophoresis tank (26.25 cm L × 15 cm W × 13.75 cm H) has three (5.7 cm L × 5.7 cm W × 1.4 cm H) liquid cold plates placed underneath for conductive cooling. Each liquid cold plate is mounted on a separate 5.7 cm L × 5.7 cm W compressible platform composed of poly-lactic acid. Two platinum electrodes in the tank generate the voltage required for electrophoresis. The placement of the electrodes is such that the electric field is aligned with the long axis of the electrophoresis chamber. Water is cooled in a second Peltier device and pumped through the liquid cold plates to maintain appropriate incubation and electrophoresis temperatures. The cooling system on the third level is similar to that on the second level—with a pump, Peltier device, and fan—but fluids within the electrophoresis tank are not subjected to direct thermoelectric cooling. The floor of the electrophoresis tank has a solenoid valve-operated central drain with a mini pump to ensure that the fluid is completely removed from the electrophoresis tank and that no residual fluid remains in the outflow pipes.

### 4.3. Comparison of the Performance of our Established High-Throughput (HT) Comet Assay and an AHT Device

The human keratinocyte cell line (HaCaT; AddexBio, San Diego, CA, USA) was used for all alkaline comet assay experiments. Cells were seeded in 12 well plates (TPP, San Diego, CA, USA) and incubated overnight at 37 °C in a 5% CO_2_ humidified incubator and a medium comprised of a 1:1 DMEM/Hams F12 medium (Quality Biological Inc, Gaithersburg, MD, USA) supplemented with 10% (*v/v*) fetal bovine serum (Gibco, Waltham, MA, USA), 1 mM sodium pyruvate (Gibco, USA), and 2 mM Glutamax (Gibco, Paisley, Scotland, UK). After removing the medium, the cells were washed with PBS, then exposed to a variety of concentrations of freshly prepared hydrogen peroxide (0–150 μM; Fisher Scientific, Fair Lawn, NJ, USA) for 30 min on ice. After the treatment, the H_2_O_2_ was removed by washing with PBS prior to analysis by HT (Cleaver Scientific, Rugby, UK), and the AHT comet assay systems. The HT comet assay method was essential, as described previously [16,23]. Briefly, 80 μL of agarose gel with a low melting point (Invitrogen, Paisley, UK), containing approximately 1.2 × 10^4^ cells) was dispensed onto glass microscope slides, pre-coated with 1% agarose with a normal melting point. The agarose was allowed to be set under a 22 × 22 mm coverslip by placing the slides on a comet assay slide chilling plate (Cleaver Scientific, Rugby, UK). The coverslips were then removed. From this point onwards, the slides were processed in parallel either in the HT system or the AHT device. The AHT device was created and programmed so that the duration, temperature, and all other buffer conditions were identical to the HT system. The slides were incubated overnight at 4 °C in a lysis buffer (2.5 M NaCl, 10 mM Tris–HCl, 100 mM disodium EDTA, pH 10 and 1% Triton X-100, all from Sigma-Aldrich, St. Louis, MO, USA) for 12 h. The slides were then washed once with double distilled water, incubated in a cold (4 °C) alkaline electrophoresis buffer (double distilled water, 300 mM NaOH, 1 mM disodium EDTA, pH ≥ 13) for 20 min and then underwent electrophoresis at 1.19 V/cm for 20 min. The slides were then incubated in a neutralization buffer (0.4 M Tris–base, pH 7.5) for 20 min, followed by washing with double distilled water. Then, the slides were stained with 2.5 μg/mL propidium iodide solution (diluted in double distilled water) for 20 min, followed by final washing and drying. At the end of the procedure, all slides were observed and scored (50 cells per gel; 100 cells per treatment) by fluorescent microscopy (Axio Scope.A1, Zeiss, Jena, Germany), and the percentage tail DNA of the comets was determined using the comet assay IV analysis software, version 4.2 (Perceptive Instruments, Haverhill, Suffolk, UK).

## 5. Patents

Drs. Cooke and Karbaschi are inventors of a patent relating to the commercially available technology used in this study. This patent is now registered with the University of South Florida [27,28].

Drs. Karbaschi, Shahm, Bhansali, and Cooke are inventors of a patent that is related to the technology that is the focus of the current work.

## Figures and Tables

**Figure 1 ijms-24-07187-f001:**
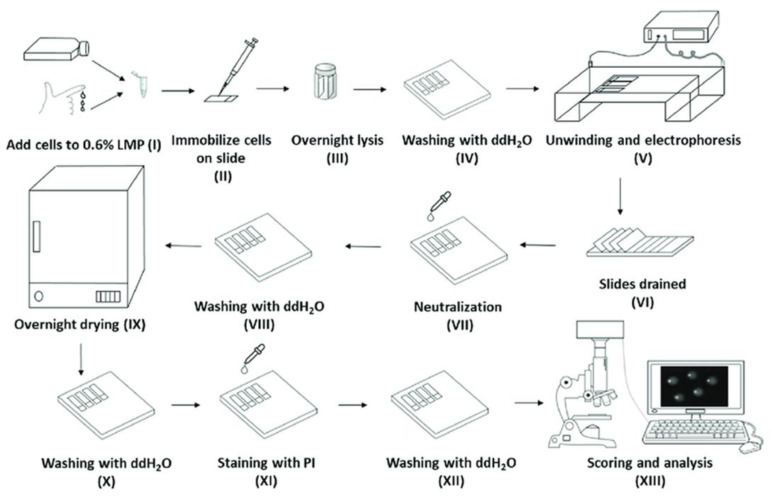
Overview of the alkaline comet assay [23]. (**I**) Cells are mixed with low-melting-point agarose prior to being (**II**) embedded onto 1% agarose pre-coated microscope slides. (**III**) The slides are then submerged in high pH lysis buffer and are then (**IV**) washed with water. (**V**) The slides are then placed in an electrophoresis buffer to allow DNA unwinding, and then undergo electrophoresis. The slides are then (**VI**) drained, (**VII**) neutralized, and (**VIII**) washed before (**IX**) drying. The slides are then (**X**) rehydrated and (**XI**) stained with a fluorescent dye, followed by another (**XII**) wash before being (**XIII**) “scored”, i.e., analyzed using fluorescent microscopy and image analysis software to quantify the damage.

**Figure 2 ijms-24-07187-f002:**
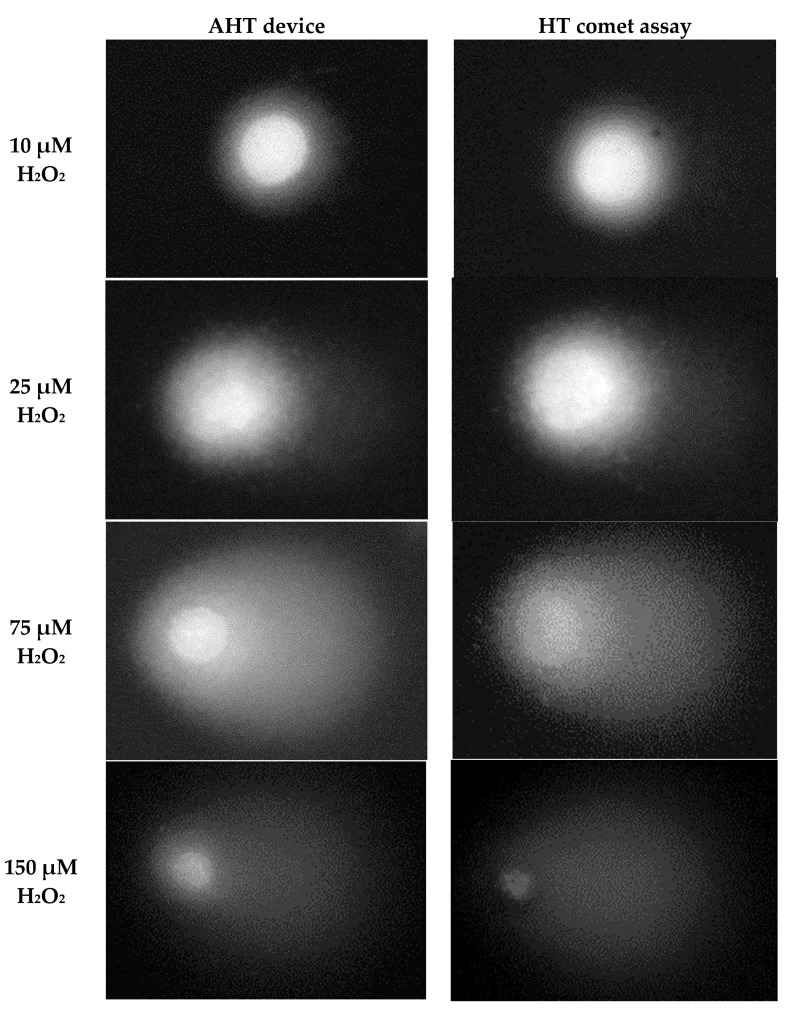
Representative images of comets generated from HaCaTs, which were incubated with 0–150 μM H_2_O_2_ prior to analysis by the HT (**right**) or AHT (**left**) approaches. The comet images generated by both assays were entirely comparable to each other in terms of the overall comet shape and regularity.

**Figure 3 ijms-24-07187-f003:**
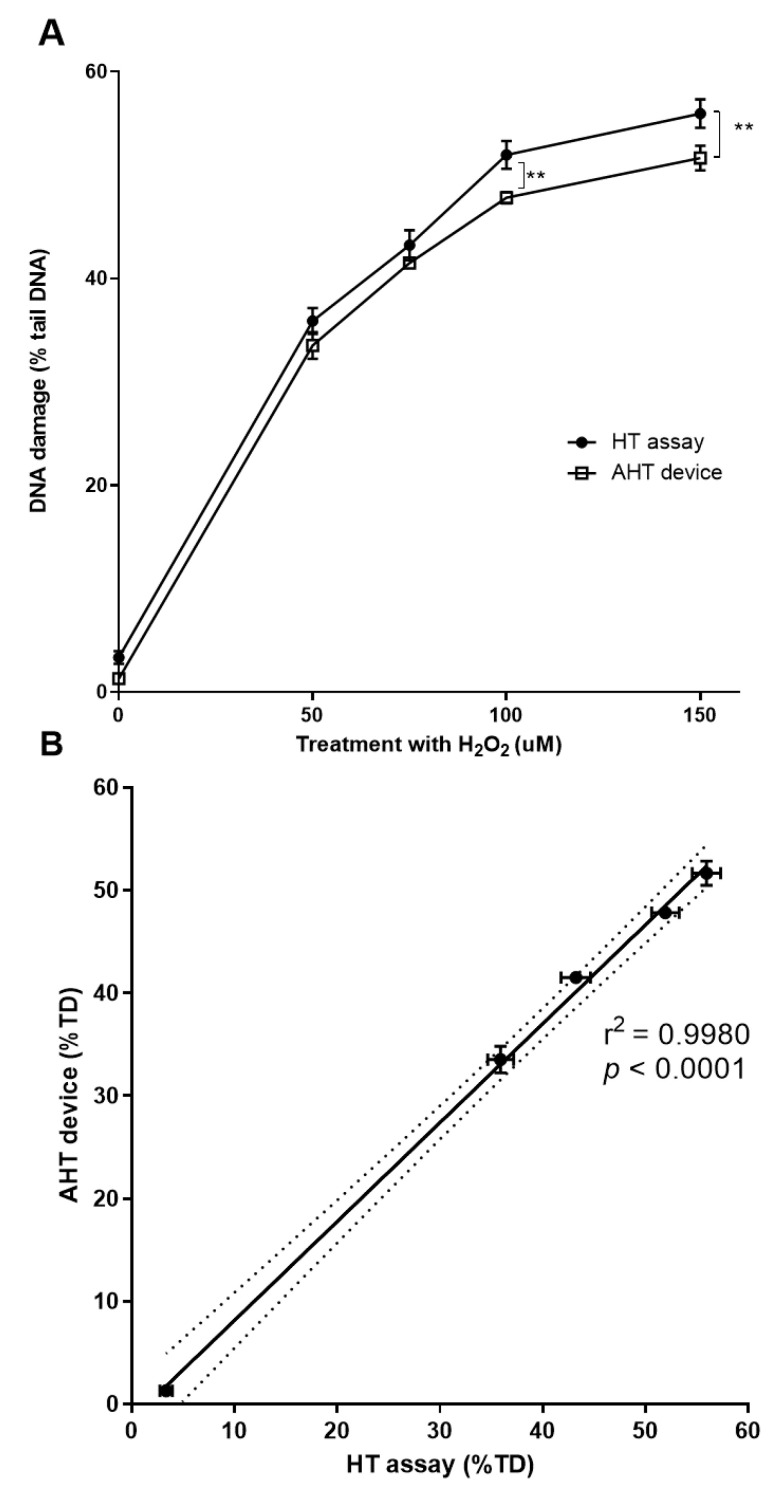
Comparison of the dose-response obtained using the HT and AHT approaches. HaCaTs were treated with 0, 50, 75, 100, and 150 µM H_2_O_2_ in duplicate, and a comet assay was performed using both HT assay and AHT device in parallel, using the same stocks of all buffers and performed on the same day. All comets, arising from both the HT assay and AHT device, were analyzed using Comet assay IV software (perceptive instruments, Bury St Edmunds, UK). Error bars represent the mean ±SEM of 100 individual determinations. (**A**) Dose-response following H_2_O_2_ treatment in samples run using either the HT (filled round symbols) or AHT (square open symbols) approach. ** represents *p* < 0.05 (**B**) Correlation of the % tail DNA of samples run using HT assay (x-axis) vs. the AHT device (y-axis). A line of best fit (solid line) was obtained using linear regression analysis using GraphPad Prism. *R*^2^ = 0.998. The range between the two dotted lines represents the corresponding 95% confidence intervals. Horizontal and vertical error bars represent the mean ±SEM of 100 individual determinations from the AHT system and HT assay, respectively.

**Figure 4 ijms-24-07187-f004:**
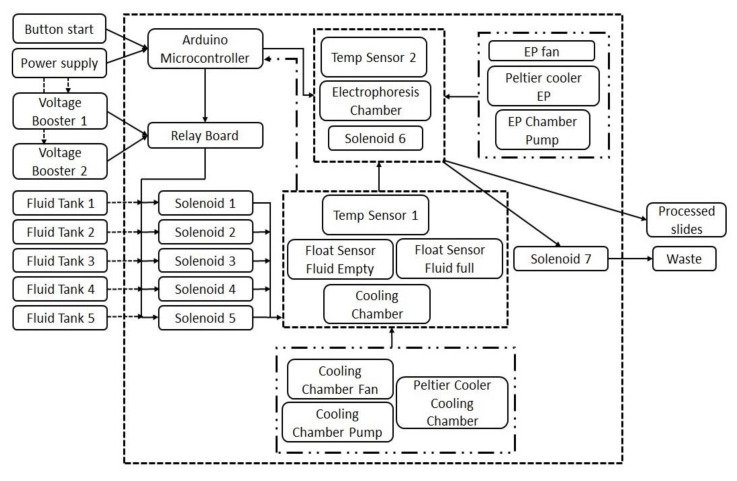
A schematic illustrating the key component of the AHT device.

**Figure 5 ijms-24-07187-f005:**
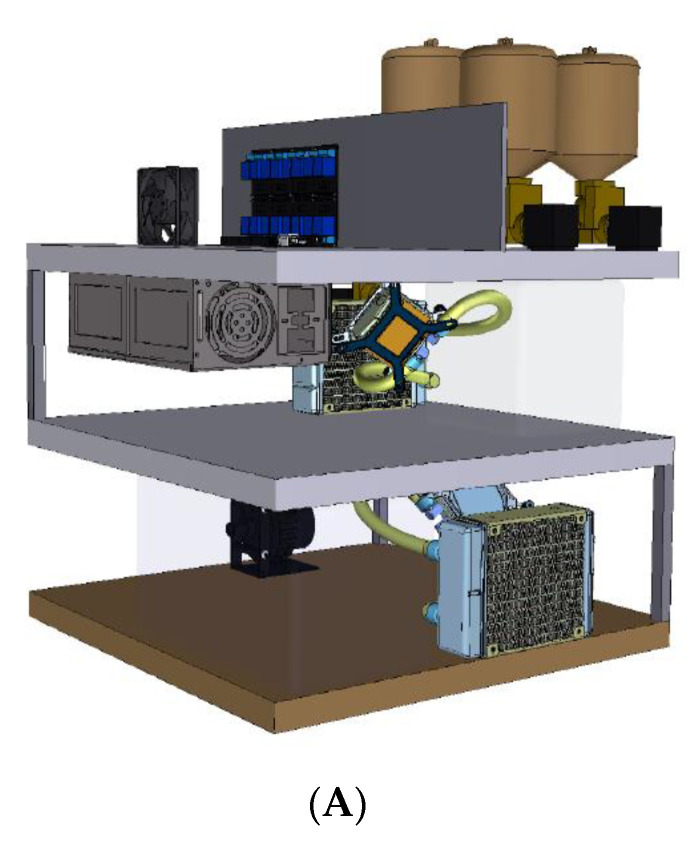
(**A**) Conceptual design and (**B**) the front view and (**C**) the back view of the operational model of the automated high-throughput comet assay device. The automated comet assay system is comprised of four levels. In Figure 5 (**A**): the top level of the device consists of solution tanks and a water tank, each with its own solenoid valve; the middle level contains a microcontroller ‘Arduino mega’; and the lowest level contains all the other electronics, and a power supply. As the process begins, the valves under the tanks open, and the liquids flow into the cooling box that is in the next level down. The next level comprises a custom-made, high-density polyethylene cooling tank and a cooling loop that combines a pump, a CPU liquid cooler to dissipate heat, and a Peltier cooler. The cooling tank is packed with two float sensors and a temperature sensor. Once the fluid is cooled to 4 °C, the cooling tank is opened, and the chilled fluid flows into the electrophoresis tank that is on the bottom level. Another cooling loop around the electrophoresis tank maintains the temperature at 4 °C throughout the process.

## Data Availability

Not applicable.

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
