# Peer review of "Development of a Novel, Automated High-Throughput Device for Performing the Comet Assay"

_ijms, 2023, doi:10.3390/ijms24087187_

Round 1
Reviewer 1 Report (New Reviewer)
The manuscript is a brief report on the newly developed Automated High-Throughput comet assay device. I read it with extreme interest due to my knowledge of the comet assay as a technique, being familiar with the difficulties in methodology since it is a sensitive procedure relying a lot on human intervention.
The new AHT device seems to remove the need for said repeated human intervention throughout the whole process, minimizing the risk of contamination. This will surely prove extremely useful in detecting specific DNA damage more accurately in future studies.
I am eagerly waiting to see results of studies using this device and employing the new methodology.
Would like to see some more important references on the comet assay technique and its evolution throughout the years (in the introduction section), but this is a brief report and I understand why it has been kept smaller in size.
Overall, the manuscript holds high scientific impact as this device can prove extremely important for researchers and laboratory technicians utilizing the comet assay.
Author Response
Response to reviewer 1
We thank the reviewer for the comments. We will include your comments for the future manuscript (more references in the introduction section). Thank you again for your time and consideration.
Reviewer 2 Report (New Reviewer)
This is a well performed study of high interest to researchers running the comet assay on a regular basis. Additional validation could have been warranted for in vivo and human samples but that is perhaps beyond the scope of the paper.
Author Response
Response to reviewer 2
We thank the reviewer for the comments. We will include in vivo and human samples studies for the future manuscript. Thank you again for your comments and consideration.
Reviewer 3 Report (New Reviewer)
The manuscript entitled “Development of a Novel, Automated High‐Throughput Device for Performing the Comet Assay” aimed to describe the development of a high throughput automated device for the comet assay based on the authors patent. Presented automated device represents a valuable, high throughput approach for reliably assessing DNA damage with minimal operator involvement, particularly if combined with automated analysis of comets which I find to be an excellent new technological achievement for the comet assay users.
Minor remarks:
Figure1 – would be nice to have it landscape orientation if possible; for the Figure 3 as well
Paragraph 2.1. – please justify the text
Please unify witting of US and/or USA
Line 343 – please present how many V/cm
Author Response
Response to reviewer 3
We thank the reviewer for the comments. I have revised the manuscript from your comments.
Minor remarks:
Figure1 – would be nice to have it landscape orientation if possible; for the Figure 3 as well.
I have finished Figure 1, but I think reviewer meant to change the Figure 4, so I have changed the Figure 4 to landscape orientation.
Paragraph 2.1. – please justify the text
I have justified the text.
Please unify witting of US and/or USA
I have changed USA to US.
Line 343 – please present how many V/cm.
I have changed 25 V to 1.19 V/cm.
Thank you again for your time and consideration.
This manuscript is a resubmission of an earlier submission. The following is a list of the peer review reports and author responses from that submission.
Round 1
Reviewer 1 Report
This paper has a purely theoretical character from the beginning till the end! I cannot see the original, experimental contribution of the authors to the proposed subject!
1. The Abstract must be seriously improved and developed! It is too short and generally described, only in theoretical meaning! The authors do not describe the most important experimental stages and results, neither the most significant conclusions derived from this study! The abstract needs serious improvements! In the abstract, authors must clearly describe the most significant and important results of the experimental completed stages and the most important conclusions!
2. The original experimental data are non-existent, are completely missing! The images from Figure 1 to Figure 5 do not suggest at all the obtained experimental results, could be copied from the literature! ! These images are not at all properly described and discussed and are completely inconclusive, in the absence of performed and adequately described tests and in the absence of tables with raw experimental data values inserted! Figure 1 , Figure 2, Figure 3 Figure 4 and Figure 5 have no scientific justification, no scientific solid experimental basis! In such cases, a proper and detailed statistical analysis is mandatory! Great Lack of experimental detailed information and experimental original proven methods and tests!!
2.1. Bibliographical references must be completely removed from the RESULTS AND DISCUSSIONS section! RESULTS AND DISCUSSIONS must describe only the original obtained results and discuss only the experimental, original work of the authors, only the experimental data obtained!! Bibliographic references should only be inserted in the general part of the Introduction possibly also in Materials and Methods section!
3. There are absolutely no pictures, graphs, neither data tables and experimental diagrams inserted into the manuscript text , that can fully support the described experimental results! The experimental data inserted into the manuscript do not properly describe at all the original findings! Where are the detailed experimental achievements and the original contributions of the authors? Graphs and diagrams drawn, that clearly highlight the described experimental results! The inserted tables have only a theoretical, general character and does not support the experimental findings! Great lack of clarity! Great lack of experimental obtained results!
4. The authors must state and describe much more experiments with obtained experimental data, tables and graphs! Figures images and graphs must be inserted and must have experimental concrete explanations based on solid and clear experimental results!
5. A statistical study is mandatory to be made for validating the obtained experimental results! In order to do this statistical analysis the authors must practically, perform the proposed experiments and obtain the necessary experimental data!
6. The aim and objectives of this experimental study are absolutely not described in the manuscript! The authors must clearly state them from the beginning!
7. This paper has a pure theoretical character only, from the beginning till the end, without any experimental scientific basis, without any experimental findings and without any conclusions derived from this study!
7.1. "Discussions" are short and general , without any significance, without any substance and clarity! Where are the experimental original authors contributions? It cannot be found nowhere! Only a theoretical description of some results, without ant scientific justification in the manuscript text!
8. Conclusions are completely missing form the end of this paper! Conclusions must be totally rewritten and reconsidered because are not at all represented in the last paragraph of this paper!! My decision is ”Reject ”. Thank you!!
Reviewer 2 Report
Dear authors.
The work submitted for review is interesting. The development of an innovative device to automate the comet test is an interesting research topic. Performing the test is labour intensive, requiring many hours of work in a dark and cool laboratory. Despite the many advantages of the test, the high labour intensity and lack of technical staff often limit its use. After reading the publication, I am not sure if this type of publication fits with IJMS. Rather, it belongs to a journal describing the metedicine of testing.
Describing new research techniques nowadays requires a large number of good quality figures. Unfortunately, the paper is prepared at best mediocre.
Key comments on the paper.
Figures 1 and 4 should be larger. Fig. 4 is difficult to read. Please supplement it with drawings.
Figure 2: The contrast of the photograph needs to be improved.
The figures should show the other parameters analysed and compare the two systems. E.g. OTM, or tail length. Describing only one parameter is insufficient - minimum 3.
Where the cells were purchased. No information available. Describe the materials used separately and then the method.
Which microscope was used for the analysis.
A flow chart describing the process of making the slides in the new system described should be completed. Similar to Fig. 1.
Round 2
Reviewer 1 Report
1. The subject approached could be interesting but the paper is very poorly structured, extremely short. 2. The experimental data obtained are extremely few, inconsistent and inconclusive and do not have no experimental support, no scientific basis proven and described in this paper. Could be copied from elsewhere, from the literature, with no experimrntal justification described! It is absolutely necessary a serious and adequate experimental justification, a proper and detailed experimental data and results interpretation and discussion! The experimental values and methods described are extremely few, very unclear presented and very inconclusive, with no experimental justification proven in the manuscript! 3. Much more additional experiments are mandatory! A statistical analysis of the final obtained results (if these results exist) is mandatory! There are only 5 figures (images), of which only two graphs are included in the same image, the rest are pictures (photos) of a very general character, that have nothing to do with the experimental part, with the experiment itself.! The pictures (images inserted) do not reflect at all and do not suggest at all the experimental methods applied and the obtained results!! Figures from Figure 1 to Figure 3 are extremely unclear and without any scientific basis! Absolutely NO tables with raw experimental data value were inserted!! NO proper, detailed and clear interpretation of the obtained results has been provided in the manuscript. The obtained experimental results are not properly and in detail described and practically are not found at all in the manuscript text!!
3..1. All the experimental part was very superficial and generally treated, without any order and without any proven scientific basis, without no significance! NO table with raw experimental data properly explained was found! There are NO experimental results processed and described in this manuscript!! Only five figures, four photos with a purely theoretical character, and two graphs without no experimental justification! Figures 1, 2 and 3 are very dark and blurry!
4. The abstract contains only four sentences without any experimental meaning, without any clarity and expression! (rest of the abstract is marked with red color and it is erased)! The abstract is practically non-existent!5. In the experimental part, all the inserted graphs, tables with data values, diagrams or images must always have a perfect experimental coverage and justification in the manuscript, based only on valid and scientifically proven experimental methods, data and values presented, which in the present work does not happen at all ! This work has a purely theoretical character, without any practical, experimental applicability., from the beginning till the end!
6. The introduction part is very short and as superficially treated as the entire work!
7. The introduction does not provide sufficient background and does not include all relevant references!
8. The research design is not appropriate and the methods are not at all properly described! The Conclusions are not at all supported by the obtained results because are completely missing in this paper! Thank you!!
Reviewer 2 Report
Dear authors, thank you for improving the manuscript and answering the questions. I wish you success in creating a solution to the comet assay.